# Physiological, Hormonal and Metabolic Responses of two Alfalfa Cultivars with Contrasting Responses to Drought

**DOI:** 10.3390/ijms20205099

**Published:** 2019-10-15

**Authors:** David Soba, Bangwei Zhou, Cesar Arrese-Igor, Sergi Munné-Bosch, Iker Aranjuelo

**Affiliations:** 1Instituto de Agrobiotecnología (IdAB), Consejo Superior de Investigaciones Científicas-Gobierno de Navarra, 31006 Mutilva, Spain; david.soba@unavarra.es; 2Key Laboratory of Vegetation Ecology, Ministry of Education, Institute of Grassland Science, Northeast Normal University, Changchun 130024, China; 9-973@163.com; 3Department of Sciences, Institute for Multidisciplinary Research in Applied Biology, Universidad Pública de Navarra, E-31006 Pamplona, Spain; cesarai@unavarra.es; 4Department of Evolutionary Biology, Ecology and Environmental Sciences, Faculty of Biology, University of Barcelona, 08028 Barcelona, Spain; smunne@ub.edu

**Keywords:** alfalfa, drought, hormones, metabolites, N/C metabolism, nodule, physiology

## Abstract

Alfalfa (*Medicago sativa* L.) is frequently constrained by environmental conditions such as drought. Within this context, it is crucial to identify the physiological and metabolic traits conferring a better performance under stressful conditions. In the current study, two alfalfa cultivars (San Isidro and Zhong Mu) with different physiological strategies were selected and subjected to water limitation conditions. Together with the physiological analyses, we proceeded to characterize the isotopic, hormone, and metabolic profiles of the different plants. According to physiological and isotopic data, Zhong Mu has a water-saver strategy, reducing water lost by closing its stomata but fixing less carbon by photosynthesis, and therefore limiting its growth under water-stressed conditions. In contrast, San Isidro has enhanced root growth to replace the water lost through transpiration due to its more open stomata, thus maintaining its biomass. Zhong Mu nodules were less able to maintain nodule N_2_ fixing activity (matching plant nitrogen (N) demand). Our data suggest that this cultivar-specific performance is linked to Asn accumulation and its consequent N-feedback nitrogenase inhibition. Additionally, we observed a hormonal reorchestration in both cultivars under drought. Therefore, our results showed an intra-specific response to drought at physiological and metabolic levels in the two alfalfa cultivars studied.

## 1. Introduction

Forage legumes are important in agricultural systems as feed sources for livestock. Among them, alfalfa (*Medicago sativa* L.) is the most cultivated forage legume in the world [1] and one of the most widely distributed forage crops from Patagonia to Siberia. The combination of high quality forage [2], the capacity to fix atmospheric nitrogen by symbiotic relationships with *Ensifer* genus bacteria, and the expected rise of forage crops due to increases in the global demand for ruminant products [3], make this crop interesting for food security and environmental conservation. However, alfalfa, like other legume crops, shows changeable production rates, mostly due to abiotic factors such as drought [4].

Drought is considered to be the most limiting factor for plant growth and crop productivity in agricultural systems [5,6]. In nitrogen fixing legumes, production limitations due to drought can be partially explained by the sensitivity of biological nitrogen fixation (BNF) to water deficit conditions [7,8]. In fact, inhibition of BNF has been shown to precede photosynthesis inhibition in drought stressed legumes [9,10]. The following mechanisms were described [11] as being responsible for inhibition of BNF in legumes: inadequate oxygen supply, carbon (C) shortage, nitrogen (N) feedback, and the role of oxidative stress. On the other hand, nodule metabolism under water restrictions is highly dependent on the metabolism of other organs [12]. For example, in alfalfa plants it was shown that nodule metabolism performance (characterized via proteomics and metabolic profile analyses) was tightly linked to the impact of drought on leaf metabolism [13]. These results showed that deleterious effects of drought on nodules also conditioned leaf responsiveness and highlighted the importance of taking into account the whole plant as a system. Nevertheless, metabolite exchange, organ signalling mechanisms, and the coordination of their changes between nodules and other plant organs are still not clear.

Abiotic stresses require plants to perceive and react to these signals in a highly coordinated and interactive manner, adapting at multiple organizational levels including morphological, physiological, and biochemical adjustments, as well as coordinating alterations in gene expression. Phytohormones play a central role in the perception of abiotic stresses, triggering interactions between the signal transduction cascades [14]. Abscisic acid (ABA) is the primary phytohormone that triggers short-term responses, such as stomatal closure, and long-term responses like favouring maintenance of root growth to optimize water uptake [15,16]. Other phytohormones such as auxin, cytokinin, gibberellins, jasmonic acid, and salicylic acid and their mutual interaction have also been suggested as being involved in abiotic stress [17,18]. At the legume nodule level, several authors have shown the importance of phytohormones in nodule formation and development (reviewed by reference [19] and more recently by reference [20]), contributing greatly to our current understanding of plant–microbe interactions. Cytokinin and local accumulation of auxin can promote nodule development [21,22,23]. Ethylene, jasmonic acid, ABA, and gibberellic acid all negatively regulate infection thread formation and nodule development [20,24]. However, in mature nodules, the factors that orchestrate nodule responses under abiotic or biotic stresses related to phytohormones have received much less attention.

Under water restriction, these changes in the level of endogenous phytohormones induce changes in gene expression that affect the synthesis, production and accumulation of several metabolites as part of an osmoregulatory response, such as carbohydrates, amino acids and organic acids, in different plant organs [25]. Several studies have clearly demonstrated that N and C metabolism plays a very important role in the water stress response [12,26,27]. It has been suggested that some amino acids can be transported from shoots to nodules as systemic signals for BNF under drought conditions [8,28] and that amino acids can represent an alternative source of C and energy for the bacteroid [29]. Further, stomatal closure and the resulting lower transpiration may involve alterations in long-distance transport of N compounds between underground and aerial plant tissues, provoking an accumulation of N compounds in nodules of plants under water stress [8]. However, it has been shown that artificial alterations of plant transpiration do not produce a decline in the rates of nitrogen fixation or an accumulation of N compounds in nodules of pea plants [30]. Therefore, it is important to understand this complex exchange of N and C metabolites between plants and nodules, especially under stress conditions.

Alfalfa is cultivated under very diverse climatic conditions around the planet, so different genotypes exposed to drought stress have revealed phenotypic variation that influences their productivity under abiotic stresses. However, the current understanding of these stress-adaptive mechanisms is relatively limited. Therefore, to reveal the key features of drought stress tolerance mechanisms in alfalfa plants, physiological, hormonal, and metabolic approaches were performed in leaves, roots, and nodules of two alfalfa cultivars with different origins and with contrasting drought responses.

## 2. Results

### 2.1. Preliminary Study

A preliminary study was conducted to choose two varieties with contrasting origins and drought responses. For this purpose, eight varieties comprising four from Spain and four from China were grown in control and drought conditions. Measures of biomass, mineral composition and gas exchange determinations were taken and are shown in Appendix A. We observed that San Isidro and Zhong Mu were the two varieties with the greatest differences in behaviour under water-stressed conditions, and therefore, these two cultivars were chosen for the study.

### 2.2. Water Limitation Effects on Plant Growth

Agro-physiological differences between cultivars are shown in Figure 1, Figure 2 and Figure 3 after suppression of irrigation until 15% of the full substrate water-holding capacity was reached. While shoot biomass decreased in the Zhong Mu under water shortage, root biomass increased in San Isidro (Figure 1a). The relative water content (RWC) was reduced in roots and, especially, in nodules for both varieties, yet the decrease in leaf water content was only significant for San Isidro (Figure 1b). The N percentage in all studied organs was reduced under water stress in both varieties (Figure 1c). The BNF efficiency under water-stressed conditions was significantly lower in both cultivars. However, drought significantly reduced the N yield in terms of biomass in Zhong Mu alone (Figure 2).

All of the gas exchange determinations: photosynthetic rates (A), transpiration (E), stomatal conductance (g_s_) and substomatic CO_2_ concentration (C_i_) were found to be significantly lower in the plants under water suppression but no differences between varieties were found in the treatments (Appendix A). Isotopic discriminations of ^13^C were analyzed in leaves and roots and a decreases in both varieties and their organs were found (Figure 3). Oxygen isotope fractions (δ^18^O) were also analyzed in leaves and roots. Differences were only found in leaves, with a significant increase in δ^18^O under water stress in Zhong Mu (Figure 3).

### 2.3. Hormones

The hormone quantitation was normalized to the dry weight (DW) of samples so as to avoid any discrepancy due to changes in relative water content under drought conditions. Three endogenous stress-related phytohormones were studied (ABA, salicylic acid (SA), and JA) (Figure 4). The ABA and JA contents were two orders of magnitude larger than the rest of the phytohormones studied. The most responsive phytohormone to drought was ABA, showing differences between treatments in all organs, but the difference between cultivars was only significant for leaves, which were the organs with the highest content. JA, which was the most abundant phytohormone, changed between treatments and cultivars only in the nodules. Their JA content decreased under water stress by 84.5% in San Isidro and by 63.3% in Zhong Mu, and the Treatment × Genotype (T × G) interaction was also significant. By contrast, SA contents increased with water shortage in leaves and roots but not in nodules with no difference between cultivars.

The four cytokinins studied are shown in Figure 5. Trans-zeatin (Z) was affected by the cultivar in roots and by drought in nodules, whereas its precursor, trans-zeatin riboside (ZR) did not differ between cultivars and was not affected by drought in leaves and nodules. There was no isopentenyl adenine (2iP) detected in nodules of alfalfa plants, and it was unaltered in roots, but, in leaves, the differences between treatments and varieties were significant. The endogenous isopentenyladenosine (IPA) content was significantly decreased in leaves and nodules under water limitation, especially in San Isidro.

The indole-3-acetic acid (IAA) content in leaves of San Isidro was higher than in Zhong Mu, especially in control conditions. The IAA content in roots of both cultivars and leaves of San Isidro was decreased under drought. Among the gibberellins analyzed, gibberellin 1 (GA_1_) was below the limit of detection in all organs studied. However, gibberellin 4 (GA_4_) showed an interesting response in nodules under water stress, being 3.5-fold higher in the nodules of both varieties under drought compared to control conditions (Figure 6).

### 2.4. Amino Acids and Sugars

Overall, 19 amino acids were quantitated in control and water-stressed plants in different plant organs. The quantitation was normalized to the dry weight (DW) of samples so as to avoid any discrepancy due to changes in relative water content under drought conditions. Significant differences in amino acids between treatments and varieties are shown in Table 1. Significant differences could be seen in amino acids involved in long-distance transport of N and osmoregulatory functions.

Three carbohydrates (glucose, fructose, and sucrose) were studied at the whole plant level (Figure 7). In leaves, the glucose concentration increased under water stress in both varieties, and fructose also increased in San Isidro but not in Zhong Mu. In contrast, the sucrose content in leaves was reduced in Zhong Mu under drought conditions. In roots under water stress, glucose was 8-fold higher in San Isidro and 4-fold higher in Zhong Mu, and the interaction between treatment and genotype was also strong (*p* = 0.000). Root sucrose content was reduced (−24%) under drought in San Isidro but unchanged in Zhong Mu. In nodules, the main sugar was sucrose, which was 2-fold higher in both cultivars under water limitation.

## 3. Discussion

During recent decades, the physiological and metabolic mechanisms underlying the responsiveness of alfalfa plants subjected to drought stress have been extensively studied in shoots. However, comparatively less information is available on other organs such as nodules. In the current study, the consequences of water restriction on growth, hormonal profile, and primary metabolism in alfalfa plants (leaves and roots) and nodules were characterized together in two cultivars with contrasting drought response strategies.

Biomass, mineral composition, and gas exchange data were used for the selection of two cultivars between a pool of 8 different cultivars from Spain and China (Appendix A). On the basis of this data, San Isidro, a cultivar grown in Spain, and Zhong Mu, from China, were chosen for the study. The cultivars were chosen for their contrasting drought response, mainly in biomass and gas exchange measures.

As observed in the preliminary study, this work identified considerable variation in the drought response within the selected varieties. Cultivars with contrasting responses to water stress are a rich resource and the study of the genetic diversity generates information to breed for increased yield under changing climate scenarios [31]. The current study aims to identify the agro-physiological, metabolite, and hormone profile parameters linked to the different responses to waters stress conditions.

### 3.1. Drought Induces Different Agrophysiological Responses between Cultivars

The study of the physiological mechanisms involved in the response to water limitation showed two different strategies in the cultivars analyzed. On the one hand, Zhong Mu maintained high RWC by closing stomata with the consequence of biomass growth inhibition. On the other hand, San Isidro maintained its biomass growth, replacing the water lost by transpiration with a more developed root system. A decrease in shoot growth is one of the most rapid effects of water shortage. It should also be noted that roots play an important role in adaptation to water limitation. When subjected to water stress, an enhanced root system has been described as improving plant water extraction from the soil [32]. As a consequence, the increase in the root/shoot ratio is a conserved response among plants exposed to water limitation [5,33], increasing the allocation of the plant’s resources from shoots to roots, reducing the evaporative surface area and improving water uptake from the soil [5,34,35]. In our study, the root/shoot ratio was enhanced in both cultivars, but each variety underwent different developmental changes to achieve this increase. In San Isidro the ratio was increased following an enhancement of the roots, whereas in Zhong Mu the aerial biomass was reduced. At the nodule level, when both varieties were considered together, nodule dry weight was slightly but significantly higher under drought; this is in accordance with the significant increase observed in alfalfa nodules under drought [12] and suggests that an increase in nodulation compensates for the lower nodule efficiency observed under this stress conditions (Figure 2).

Maintenance of a constant RWC by closing stomata has been described as another physiological mechanism in drought-tolerant cultivars in many legumes and other crops [36]. However, our data showed a significant decrease in water content in all of the organs studied in both cultivars under drought, without differences between cultivars. Carbon isotope discrimination (∆^13^C) has been described as a good indicator of water status, providing long-term information on the transpiration efficiency of plants and as an indicator of water use efficiency (WUE) [37,38,39]. Given that leaves and roots of Zhong Mu maintained lower ∆^13^C under water-stressed conditions, we could assume that this cultivar had lower transpiration and greater WUE during biomass formation compared to San Isidro, and this was in accordance with the observed leaf RWC. These leaf RWC and ∆^13^C results were also in agreement with the oxygen isotope ratio (δ^18^O) (Figure 3). Some authors [40,41,42,43] have suggested that the δ^18^O of plant material reflects the evaporative conditions under which the material was formed, showing a negative relationship between the δ^18^O of leaves and stomatal conductance. In our experiment, the δ^18^O of leaves showed increased values under water limitation in both cultivars but it was only statistically significant in Zhong Mu. In line with the authors cited previously, this data could indicate lower stomatal conductance in Zhong Mu leaves.

Therefore, taking the physiological and isotopic data together, we observed two strategies to deal with water deficit. While San Isidro fixed more carbon, maintaining its aerial biomass under drought and promoting its root system for increased water uptake from the soil to compensate for water lost by transpiration, Zhong Mu reduced its water loss through transpiration by closing stomata and therefore fixed less carbon. Consequently, we have in both cultivars a trade-off between saving water and capturing carbon.

### 3.2. Drought Caused Hormonal Reorchestration at the Whole Plant Level

Under environmental stresses like drought, plants respond by changing the levels of endogenous phytohormones, playing a central role in the rapid orchestration of the stress response. ABA has been proven as the main phytohormone involved in abiotic stresses, provoking the rapid closure of stomata after perception of stress [44]. After water deprivation, the endogenous levels of ABA in leaves were three-fold higher in San Isidro and four-fold higher in Zhong Mu. These results are in agreement with the isotopic data that suggested a lower stomatal conductance in Zhong Mu plants exposed to drought. Roots under drought also produce ABA, which is rapidly transported from roots to shoots where it is accumulated in the apoplast of leaf guard cells [14,45,46]. The lower ABA content observed in roots compared with leaves indicates efficient ABA transport from roots to shoots. At the nodule level, in *Pisum sativum* it has been shown that an exogenous ABA supply stimulated an abrupt decrease in the nitrogenase activity of nodules, caused by the decrease in leghemoglobin levels [47]. Thus, the observed increase in ABA content under drought might have an effect on nodule oxygen diffusion and lead to the decline in nitrogen fixation.

Salicylic acid is also accumulated in plants under drought stress and is involved in inducing drought tolerance by regulating several physiological processes through signalling, for example, leaf senescence [48]. Our data showed a small accumulation of salicylic acid under drought in leaves and roots, although without differences between the cultivars. In contrast, jasmonic acid was strongly reduced in nodules under drought with the greater reduction observed for San Isidro. Jasmonic acid has been reported to be a negative regulator in the initial moments of nodulation [49,50]. Our results could suggest a cultivar-specific enhancement of nodulation under water limitation that compensates for the decrease in nitrogenase activity.

In San Isidro nodules a significant decrease was found in the cytokinins, zeatin, and IPA, implying a cultivar-specific trend for these two hormones. Recently, several authors have speculated about a direct connection between cytokinins and nitrogen fixation metabolism [51,52,53]. Additionally, it has been demonstrated that a decreased level of cytokinins accelerates nodule senescence and thus causes a gradual decline in nitrogenase activity throughout nodule development [52]. Cytokinins are often considered ABA antagonists [54,55], so a reduction in cytokinins might amplify responses to increasing levels of ABA [56]. Therefore, the observed reduction in cytokinins in San Isidro nodules may be in accordance with the increased levels of ABA under drought, reducing the nitrogen fixation efficiency in this cultivar, and suggesting that a hormonal cross-talk coordinates the drought response.

It has been reported that reduced levels of gibberellins decrease the number of nodules and this suggests that nodule formation is strictly controlled by the gibberellin concentration [24,57]. Our observed four-fold increase in gibberellin contents in nodules under drought conditions might involve an enhancement of nodule formation in both varieties, as was observed for jasmonic acid. This is in accordance with the observed enhancement of nodule biomass under water-stressed conditions.

### 3.3. Drought Induces Lower N_2_ Fixation, and Cultivar Differences could be Explained by N Feedback

The decrease in %N indicates that, under water limitation, the nodules (all plant N was provided by the nodules) were not efficient enough to supply the required N and to solve the N availability problems. The strong relationship between N_2_ fixation in legumes and the physiological status of the host plant has been described in several studies [12,26,58,59,60,61,62]. The greater decrease of N biomass yield and BNF efficiency between treatments in Zhong Mu showed a more limiting nodule functioning in this cultivar. Deleterious effects of drought on nodule performance have been described previously in alfalfa, and several mechanisms have been suggested for the inhibition of nitrogen fixation in legumes: carbon shortage, nitrogen feedback, inadequate oxygen supply and the role of oxidative stress [11,12,13,27,61,63].

Sucrose is the principal form of photoassimilates for long-distance transport. Leaf sucrose concentration is determined by several factors including the rate of photosynthesis, the rate of sucrose hydrolysis (to glucose and fructose), and the rate of sucrose export to other organs [64]. In our experiment, the decrease in leaf sucrose concentration between treatments was only significant in Zhong Mu. Conversely, larger differences between treatments in glucose and fructose content in leaves were found in San Isidro, which suggested a lower rate of photosynthesis (as shown before) or/and higher rate of sucrose export to other organs in Zhong Mu plants (Figure 7). Contrastingly, the root sucrose decrease in San Isidro might be related with the larger increase in root glucose content or with the greater (but not significant) accumulation of sucrose in nodules of this cultivar. The accumulation of sucrose under drought stress has been widely reported in nodules of alfalfa and other legumes and was mainly attributed to the inhibition of SS activity [60,65]. Our data showed a similar accumulation of sucrose in nodules under water shortage in both cultivars, indicating that sucrose accumulation in nodules is not a good measure to distinguish drought tolerance between genotypes.

Leaf nitrogenous metabolites (Asn, Asp, Glu) were decreased by water limitation, probably due to the inhibition of nitrogenase activity (Table 1 and Figure 2). The depletion at the leaf level of these amino acids suggests that under drought conditions there was a remobilisation of N from the leaf towards reserve organs, mainly the primary root [13,66]. The accumulation of Pro might serve to stabilize the protein structure and is associated with an osmoregulatory function in response to drought [67]. High levels of proline in nodules have been observed [18] to have a protective role against reactive oxygen species (ROS). Within this context, the increased levels registered in drought-stressed plants suggest that these nodules could have been subjected to oxidative stress. Accumulation of ROS is another mechanism that might play a role in the drought-induced inhibition of N_2_ fixation. It should be highlighted that Pro accumulation under drought has been described as an important characteristic for distinguishing drought tolerance between cultivars [68]. Nevertheless, in our work the enhancement of Pro under water shortage was similar in both cultivars, indicating that Pro is not the prime cause for distinct drought tolerance in either cultivar, as similarly observed in nodules of two contrasting drought-tolerant peanut cultivars [33]. In addition, gamma-aminobutyric acid (GABA) is an amino acid with a protective function against ROS, and which acts during osmoregulation and as a signalling molecule [69]. The accumulations of GABA in nodules and roots of San Isidro plants were larger than in Zhong Mu.

Asn is the main N-transporter in alfalfa, and Asn accumulation in nodules has been implicated in N feedback, promoting the inhibition of symbiotic N_2_ fixation [8,27,28]. According to these studies, accumulation of Asn and other N_2_ fixation products in nodules is due to the lower aboveground N demand. In leaves, the Asn content was significantly lower under water limitation in San Isidro but not in Zhong Mu. Such differences suggest a higher consumption of Asn in San Isidro that resulted in a higher demand and lower nodule accumulation of Asn compared to Zhong Mu. This accumulation of Asn in Zhong Mu nodules could have been involved in the observed lower N_2_ fixation efficiency. Additionally, another reason to explain the greater accumulation of Asn in Zhong Mu nodules is its lower transpiration rate, as observed previously with the ∆^13^C and δ^18^O data. It has been noted [8] that lower transpiration may involve alterations in long-distance transport of N compounds between underground and aerial plant tissues, provoking an accumulation of N compounds in nodules of plants under water stress. However, recently, examination of the influence of transpiration on long-distance metabolite transport and its effects on the drought-induced inhibition of symbiotic nitrogen fixation in pea nodules under artificial reduction of plant transpiration has not revealed any accumulation of N compounds [30].

## 4. Materials and Methods

### 4.1. Plant Material and Experimental Design

The experiments were conducted with two alfalfa (*Medicago sativa* L.) cultivars, San Isidro and Zhong Mu, currently cultivated in Spain and China, respectively. These two contrasting cultivars were chosen during a previous experiment between eight alfalfa cultivars (four from Spain and four from China). The design was similar in both experiments. Seeds were germinated in Petri dishes and after germination, the plants were transplanted to 6 L black pots (four plants per pot) containing a substrate filled with 2:1 (*v*/*v*) perlite/vermiculite. The experiment was conducted with six pots of each combination in a controlled greenhouse at 25/18·°C(day/night) under natural daylight. The greenhouse was located at the Institute of Agrobiotechnology (IdAB) (42°47′N, 1°37′W; Pamplona, Spain).

During the second week, plants were inoculated twice with *Ensifer meliloti* strain 102F34. To ensure that the sole N source was N_2_ fixed by nodules, the plants were watered with an N-free nutrient solution with the following composition (mEq/L): KH_2_PO_4_, 2; MgSO_4_, 1.5; KCl, 3; CaCl_2_, 1.5; Na-FeEDTA, 0.12; and microelements as recommended [70], twice a week. When plants were 62 days old, half of the plants (randomly selected) were exposed to drought conditions by withholding watering), while the others were maintained in optimal water conditions. Suppression of irrigation was maintained for 15 days until 15% of the full substrate water-holding capacity was reached, while control plants were maintained at 90% full substrate water-holding capacity. At this point, gas exchange measurements were carried out, and then plants were collected and separated into apical shoot, primary root and nodules and were immediately frozen in liquid N and stored at −80 °C for further analyses. A subsample of each organ was separated and dried in an oven for 48 h at 60 °C in order to determine dry weight. The water status of the plants was evaluated by measuring the leaf, root and nodule RWC according to [71].

Biological N fixation efficiency (BNF efficiency) was calculated as the quotient between biomass N yield per plant and nodule dry matter per plant.

### 4.2. Gas Exchange and Chlorophyll Fluorescence Determinations

Gas exchange measurements were carried out with a Li-Cor 6400 portable gas exchange photosynthesis system (LI-COR, Lincoln, NE, USA) on healthy and fully expanded apical leaves under conditions similar to growth conditions (400 μmol m^−2^ s^−1^ PPFD, 25 °C). Photosynthetic CO_2_ assimilation (A) was determined using equations developed by reference [72]. Stomatal conductance (gs) was determined as described by reference [73].

### 4.3. Hormone Profiling

Leaf, root, and nodule samples were ground in liquid nitrogen using a mix ball and 100 mg sample (fresh weight) was homogenised with methanol:isopropanol:acetonitrile (50:49:1) using ultrasonication for 30 min (4 °C). After centrifuging at 14,000× *g* for 10 min at 4 °C, the supernatant was collected and the pellet re-extracted with the same solvent until it was colourless. Supernatants were pooled and filtered with 0.22 μm PTFE filters (Phenomenex, Torrance, CA, USA), transferred to HPLC vials and injected into a UHPLC-MS/MS. Endogenous hormones including abscisic acid (ABA), salicylic acid (SA), jasmonic acid (JA), the auxin indole-3-acetic acid (IAA), the cytokinins trans-zeatin and trans-zeatin riboside, isopentenyl adenosine (IPA) and 2-isopentenyl adenine (2iP), and gibberellin 4 (GA_4_), were separated using an elution gradient on a reverse-phase UHPLC system and quantified using tandem mass spectrometry in multiple reaction monitoring mode exactly as described [74]. UHPLC/ESI-MS/MS equipment with an Aquity UPLCTM Sistem (Waters, Milford, MA, USA) quaternary pump equipped with and autosampler coupled to an API 3000 triple quadrupole mass spectrometer (PE Sciex, Concord, Ont., Canada) was used for analyses. Deuterium-labelled compounds were used as internal standards for quantification.

### 4.4. Free Amino Acid and Sugar Determinations

For the free amino acid determination, frozen plant tissues were ground to a fine powder in liquid nitrogen and a sub-sample was lyophilised. Lyophilised plant tissue (20 mg) was homogenised in 400 μL of 80% ethanol and mixed using a vortex, incubated at 80 °C for 1 h, and centrifuged at 14,000× *g* and 4 °C for 10 min and the pellet was completely dehydrated. The pellet was re-suspended in 100 μL of milli-Q water, centrifuged at 14,000× *g* and 4 °C for 10 min, and the supernatant was collected. The amino acid content in the supernatant was determined by high performance liquid chromatography (HPLC) (Waters Corporation, Barcelona, Spain) after derivatization with a ACCQ-Fluor™ Reagent kit (Waters, Milford, MA, USA) based in borate buffer, acetonitrile and, AQC derivatizing reagent (6-aminoquinolyl-n-hydroxysuccinimidyl carbamate) as previously described [75].

For soluble sugars determinations, lyophilised plant tissue (25 mg) was homogenised in 0.5 mL of 100% ethanol, then another 0.5 mL of 80% ethanol was added and mixed using a vortex. The sample was incubated at 70 °C for 90 min, centrifuged at 14,000× *g* for 10 min and the supernatant collected. The supernatant was used to determine glucose, fructose, and sucrose content with an ionic chromatographer (ICS-3000, Thermo Scientific™, Waltham, MA, USA).

### 4.5. Carbon and Nitrogen Content

The C and N content in leaf, root and nodule samples were determined based on sample dynamic combustion, using an elemental analyzer (FlashEA1112, ThermoFinnigan, Waltham, MA, USA) equipped with a MAS200R autosampler. The sample was weighed in a tin capsule (MX5 microbalance, Mettler-Toledo, Columbus, OH, USA) and introduced into a quartz reactor filled with WO3 and copper and heated at 1020 °C. The combustion gas mixture was carried by a helium flow to a WO3 layer to achieve a complete quantitative oxidation, following by a reduction step in a copper layer to reduce nitrogen oxides and SO_3_ to N_2_ and SO_2_. The resulting components, N_2_, CO_2_, H_2_O, and SO_2_ were separated in a chromatographic column (Porapak 2m, Santa Clara, CA, USA) and detected with a thermal conductivity detector.

### 4.6. Carbon ^13^C Discrimination and Oxygen Isotopic Composition Analyses (Δ^13^C and δ^18^O Respectively)

Leaf and root samples (≈0.1 g) were selected to determinate the C isotopic composition (δ^13^C) using an elemental analyser (EA1108; Carlo Erba Instrumentazione, Milan, Italia) coupled to an isotope ratio mass spectrometer (Delta C; Finnigan, Mat., Bremen, Germany) operating in continuous flow mode. The δ^13^C values were transformed to discrimination values (Δ^13^C) according to [38], where air δ^13^C = −8‰ in Vienna Pee Dee Belemnite (V-PDB):
(1)$$\Delta^{13}C=\frac{\delta \;air-\delta \;plant}{1-\frac{\delta \;plant}{1000}}$$


A MAT253 isotope ratio mass spectrometer coupled through a Conflo III to a high temperature conversion/elemental analyser (TC/EA) (all of them from Thermo Scientific™, Waltham, MA, USA) was used for ^18^O/^16^O isotope ratio measurement. Samples were weighed out (0.6 mg) into 5 × 3.3 mm silver capsules (Lüdi, Switzerland). Samples were thermally decomposed to CO in a glassy carbon reactor. The reactor consisted of an outer ceramic tube and an inner glassy carbon tube filled with glassy carbon granulate, and silver, and quartz wool (Elemental Microanalysis). The reactor temperature was set to 1450 °C and the post-reactor gas chromatography (GC) column was maintained at 65 °C (He flow rate 90 mL/min). The ^18^O/^16^O isotope ratios are reported as δ^18^O values (%) relative to Vienna Standard Mean Ocean Water (V-SMOW). The measured δ^18^O values were normalized to the VSMOW scale using the contemporaneously analyzed standards IAEA-601 (δ^18^O_VSMOW_=23.3 ± 0.3‰) and IAEA-602(δ^18^O_VSMOW_ = 71.4 ± 0.5 ‰). The SD for measurements was ± 0.4 ‰.

### 4.7. Statistics

Statistical analyses were performed with IBM SPSS Statistics for Windows, Version 20.0. (IBM Corp. Armonk, NY, USA). Differences among well-watered and drought treatments were evaluated with two way Analyses of Variance (ANOVA), with treatment being one fixed factor and variety the other fixed factor. Tukey’ post hoc tests were used to determine statistical differences between treatments and varieties. All data were tested for normality (Kolmogorov–Smirnof test) and homogeneity of variances (Levene’s test). The resulting P-values were considered to be statistically significant at *p* < 0.05. Asterisks indicate significant differences: * *p* < 0.05, ** *p* < 0.01, *** *p*< 0.001, in the two-way ANOVA for water-stress and cultivars.

## 5. Conclusions

Plant response to water limitation is a trade-off between saving water by reducing transpiration and capturing carbon through photosynthesis. This response could be different between genotypes within the same crop species. In our study, two different strategies were observed in two alfalfa cultivars selected in a previous experiment. According to physiological and isotopic data, Zhong Mu has a water-saver strategy via a more conservative response: reducing of water loss by closing stomata and fixing less carbon by photosynthesis, thus limiting its growth under water limitation. In contrast, San Isidro promoted root growth to enhance water uptake to replace water lost through transpiration due to its more open stomata, allowing a greater rate of carbon to be fixed and maintaining biomass under drought. ABA was found in higher concentrations under drought in Zhong Mu leaves than in San Isidro leaves, supporting the physiological and isotopic data. The root nodule response to water deficit was also different between cultivars. Data on the biomass N yield and BNF efficiency indicated a lower capacity to supply the required N in Zhong Mu. Our data suggest that this cultivar-specific decrease in nitrogen fixation is caused by a nitrogenase inhibition due to an N-feedback, which was manifested in the accumulation of Asn in nodules exposed to water shortage. On the other hand, the accumulation of sucrose, proline, and ABA in nodules of both cultivars might implicate carbon shortage, oxidative stress, and inadequate oxygen supply to the bacteroids, thereby reducing the N fixed.

## Figures and Tables

**Figure 1 ijms-20-05099-f001:**
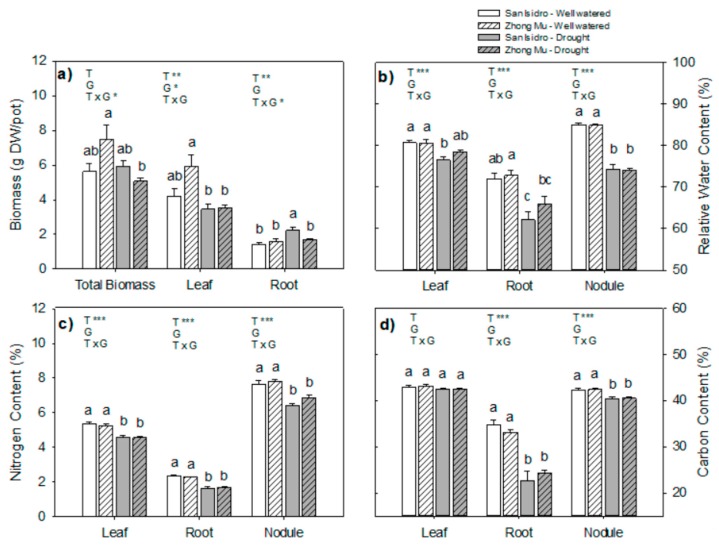
Agro-physiological parameters under control and drought conditions in two cultivars of Medicago sativa: San Isidro and Zhong Mu. (**a**) Dry biomass (g DW/pot); (**b**) relative water content (%); (**c**) nitrogen content (%) and (**d**) carbon content (%). Each value represents the mean ± SE (*n* = 6). The different letters indicate significant differences (*p* < 0.05). Asterisks indicate significant differences: * *p* < 0.05, ** *p* < 0.01, *** *p*< 0.001, in the two-way ANOVA for water-stress and cultivars.

**Figure 2 ijms-20-05099-f002:**
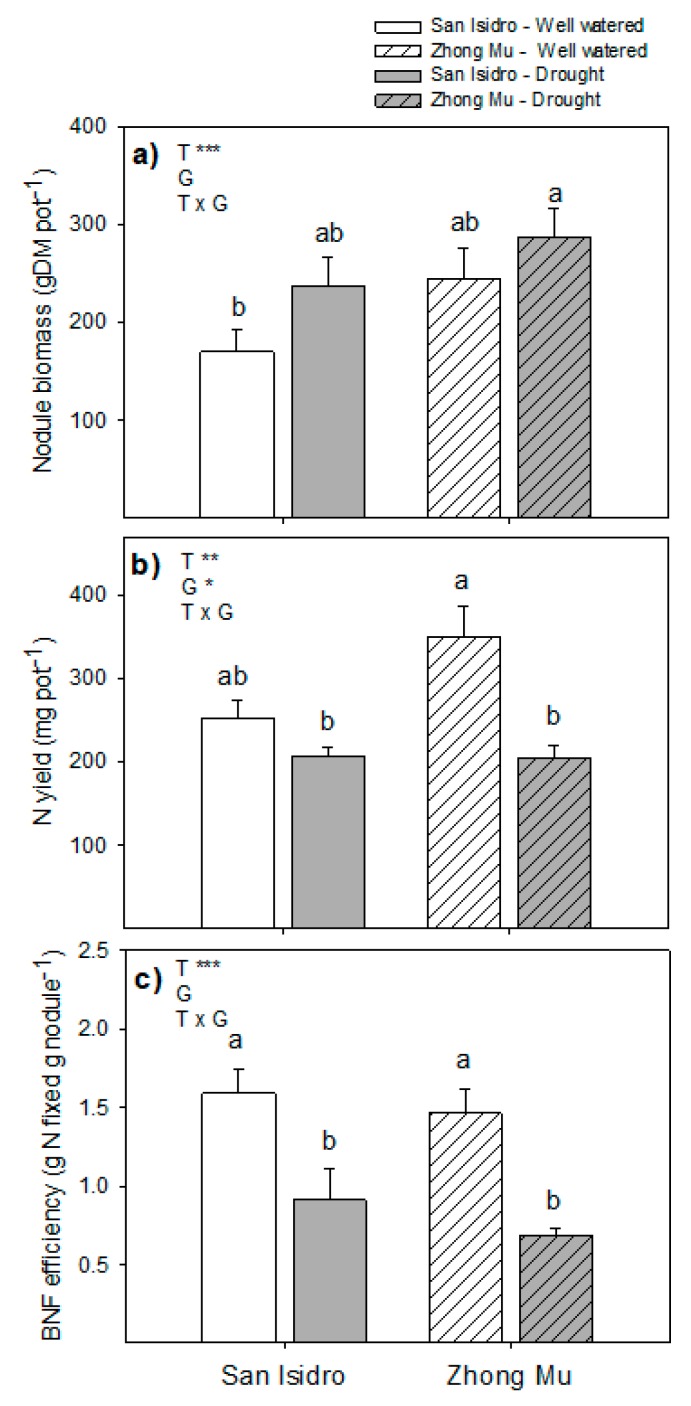
(**a**) Nodule biomass; (**b**) N yield per pot and (**c**) biological nitrogen fixation (BNF) efficiency under control and drought conditions in two cultivars of *Medicago sativa*: San Isidro and Zhong Mu. Each value represents the mean ± SE (*n* = 6). The different letters indicate significant differences (*p* < 0.05). Asterisks indicate significant differences: * *p* < 0.05, ** *p* < 0.01, *** *p*< 0.001, in the two-way ANOVA for water-stress and cultivars.

**Figure 3 ijms-20-05099-f003:**
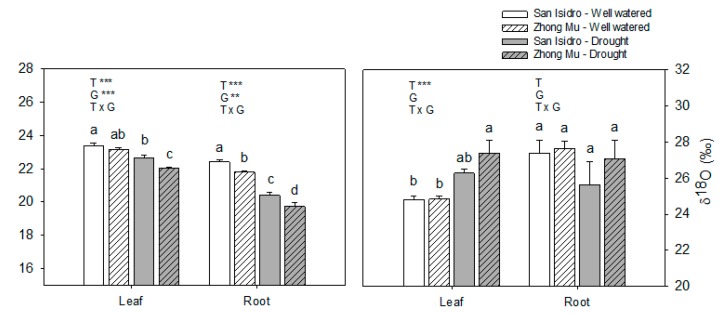
Carbon isotope discrimination (∆^13^C) and oxygen isotope composition (δ^18^O) in leaves and roots under control and drought condition in two cultivars of *Medicago sativa*: San Isidro and Zhong Mu. Each value represents the mean ± SE (*n* = 6). The different letters indicate significant differences (*p* < 0.05). Asterisks indicate significant differences: * *p* < 0.05, ** *p* < 0.01, *** *p*< 0.001, in the two-way ANOVA for water-stress and cultivars.

**Figure 4 ijms-20-05099-f004:**
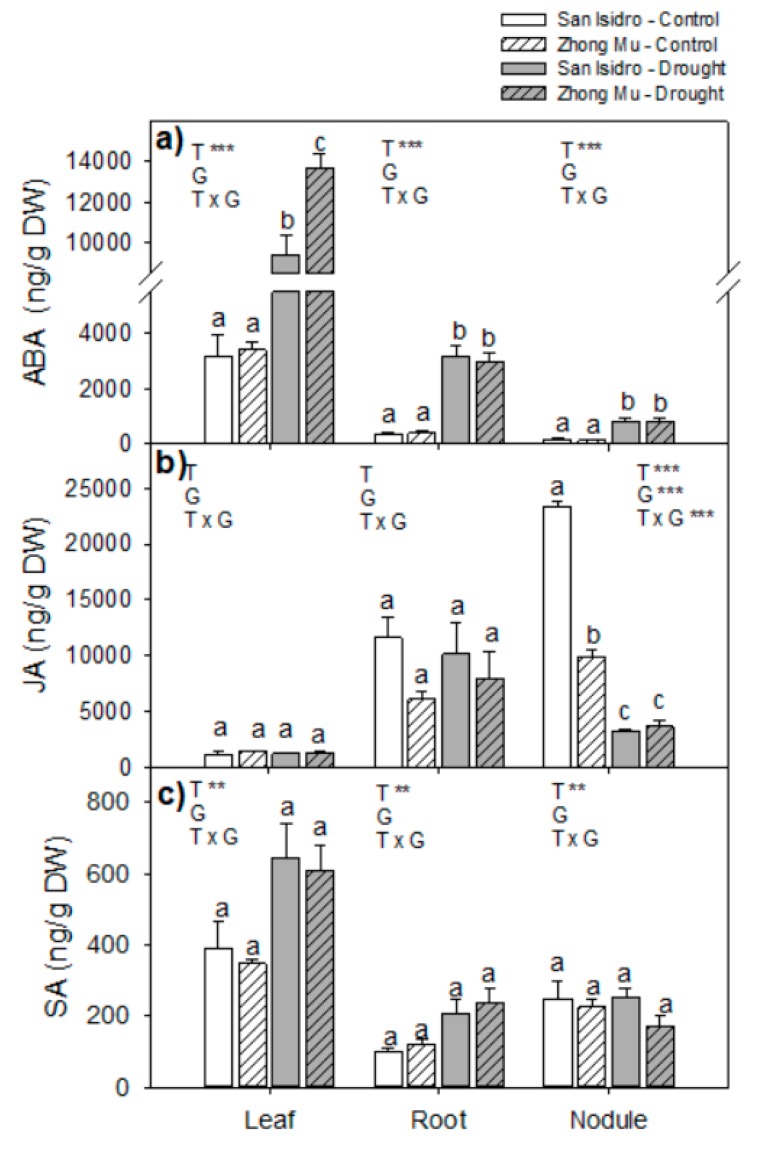
Stresses related hormones: (**a**) Abscisic acid (ABA); (**b**) jasmonic acid (JA) and (**c**) salicylic acid (SA) content in leaves, roots and nodules under control and drought condition in two cultivars of *Medicago sativa*: San Isidro and Zhong Mu. Each value represents the mean ± SE (*n* = 6). The different letters indicate significant differences (*p* < 0.05). Asterisks indicate significant differences: * *p* < 0.05, ** *p* < 0.01, *** *p*< 0.001, in the two-way ANOVA for water-stress and cultivars.

**Figure 5 ijms-20-05099-f005:**
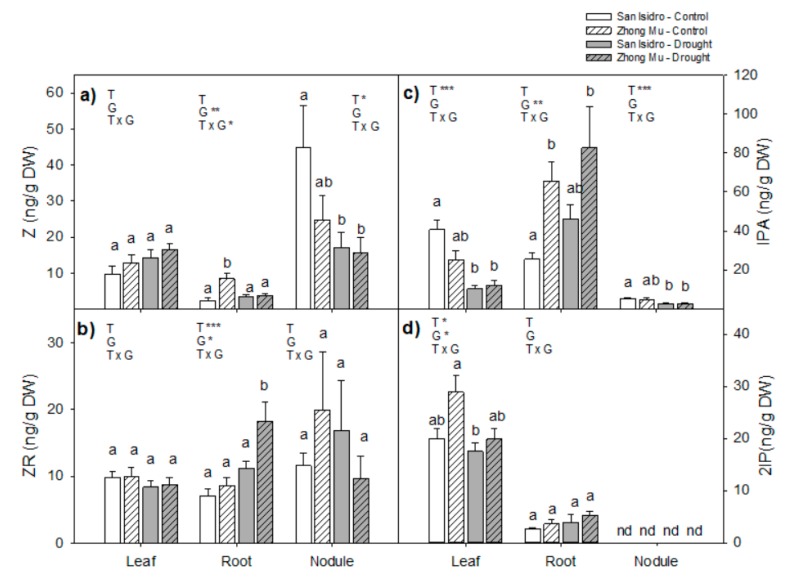
Cytokinins content in leaves, roots and nodules under control and drought condition in two cultivars of *Medicago sativa*: San Isidro and Zhong Mu: (**a**) trans-zeatin (Z), (**b**) trans-zeatin riboside (ZR), (**c**) isopentenyladenosine (IPA) and (**d**) isopentenyl adenine (2iP). Each value represents the mean ± SE (*n* = 6). The different letters indicate significant differences (*p* < 0.05). Asterisks indicate significant differences: * *p* < 0.05, ** *p* < 0.01, *** *p*< 0.001, in the two-way ANOVA for water-stress and cultivars.

**Figure 6 ijms-20-05099-f006:**
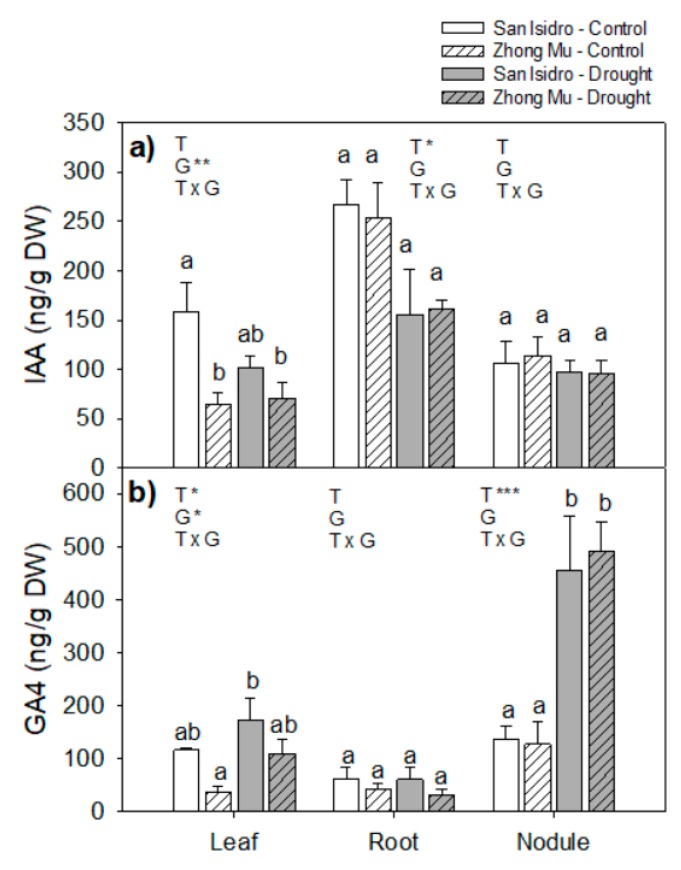
(**a**) Indole-3-acetic acid acid (IAA) and (**b**) gibberellin 4 (GA_4_) content in leaves, roots and nodules under control and drought condition in two cultivars of *Medicago sativa*: San Isidro and Zhong Mu. Each value represents the mean ± SE (*n* = 6). The different letters indicate significant differences (*p* < 0.05). Asterisks indicate significant differences: * *p* < 0.05, ** *p* < 0.01, *** *p*< 0.001, in the two-way ANOVA for water-stress and cultivars.

**Figure 7 ijms-20-05099-f007:**
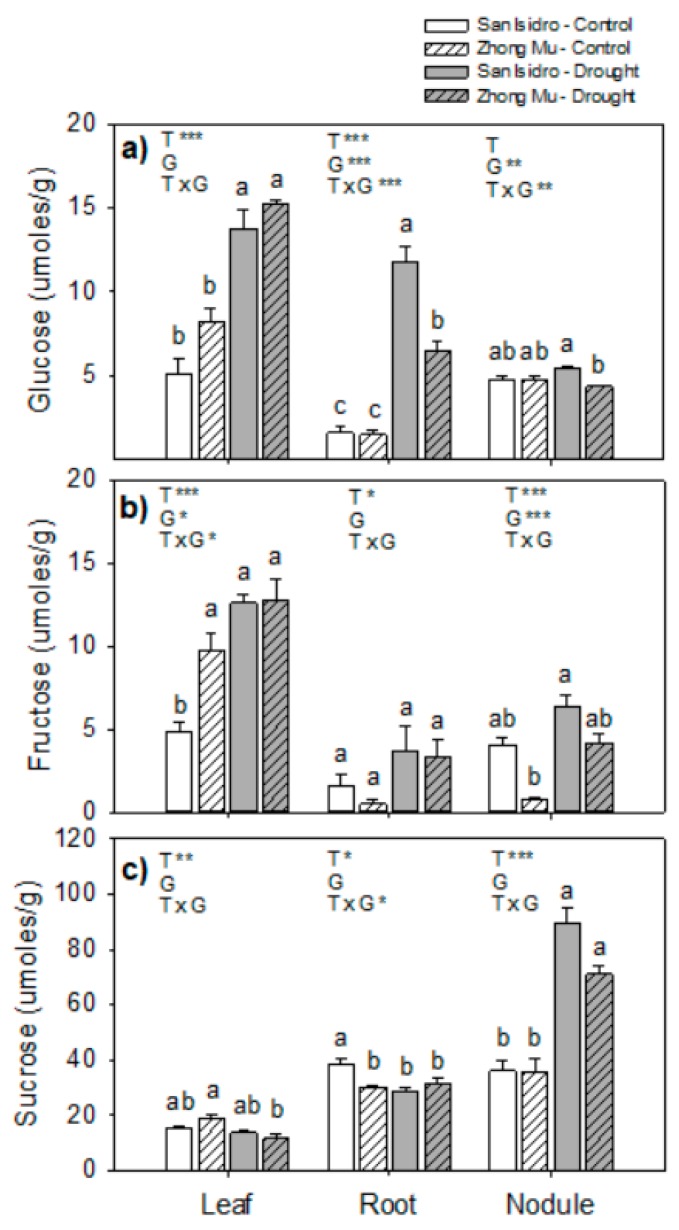
Soluble sugars content in leaves, roots and nodules under control and drought condition in two cultivars of *Medicago sativa*: San Isidro and Zhong Mu: (**a**) glucose, (**b**) fructose, (**c**) sucrose. Each value represents the mean ± SE (*n* = 6). The different letters indicate significant differences (*p* < 0.05). Asterisks indicate significant differences: * *p* < 0.05, ** *p* < 0.01, *** *p*< 0.001, in the two-way ANOVA for water-stress and cultivars.

**Table 1 ijms-20-05099-t001:** Amino acid quantification (nmol/g DW) in leaves, roots and nodules of *Medicago sativa* under well-watered (WW) and water-stressed conditions (WS) determined by HPLC.

	Leaves				Root				Nodule			
	San Isidro		Zhong Mu		San Isidro		Zhong Mu		San Isidro		Zhong Mu	
	WW	WS	WW	WS	WW	WS	WW	WS	WW	WS	WW	WS
Asp	0.76 b	0.45 c	0.97 a	0.71 b	0.86 a	0.55 b	0.69 b	0.60 b	1.07	1.24	1.31	1.01
Glu	2.05 a	0.92 b	2.37 a	1.03 b	1.23 a	0.75 c	0.97 ab	0.86 bc	4.56	3.70	4.49	3.41
Ser	1.51	1.65	1.73	1.56	0.98	0.84	0.83	0.76	0.84 b	1.55 a	0.97 b	1.12 b
Asn	9.88 a	3.62 c	7.80 b	5.07 bc	56.92 b	44.20 b	92.02 a	59.29 b	138.73 ab	150.90 ab	108.41 b	179.55 a
Gly	0.50	0.59	0.58	0.62	1.39 a	0.79 b	0.99 b	0.84 b	0.81 ab	1.01 a	0.72 b	0.84 ab
Gln	0.62	0.60	0.62	0.68	0.70	0.88	0.68	0.67	0.81	1.20	1.13	1.19
His	0.24 ab	0.26 ab	0.21 b	0.30 a	0.21	0.23	0.22	0.21	0.38 b	1.06 a	0.50 b	0.85 a
Thr	0.97	1.60	1.51	2.03	0.93	1.25	0.88	1.11	0.77 b	1.43 a	0.75 b	0.95 b
Ala	27.16 a	23.80 b	23.43 b	24.54 b	8.73	8.34	10.79	10.00	19.75	19.05	21.42	26.16
Arg	1.26	1.82	1.40	1.98	1.20	1.46	0.85	1.28	0.51 bc	1.05 a	0.48 c	0.85 ab
GABA	1.16	1.68	1.29	1.77	1.03 c	2.85 a	0.79 c	1.83 b	1.25 b	3.29 a	1.11 b	2.56 a
Pro	1.47 b	5.83 a	1.56 b	5.64 a	3.16 b	9.98 a	3.67 b	12.40 a	1.80 b	18.93 a	2.60 b	19.86 a
Tyr	0.55	0.60	0.53	0.69	0.36 b	0.43 a	0.37 b	0.38 b	0.56 c	1.05 a	0.54 c	0.81 b
Val	0.56 b	0.78 ab	0.56 b	0.91 b	0.31 c	0.62 a	0.32 c	0.54 b	0.53 b	1.70 a	0.54 b	1.24 a
Met	0.40 ab	0.46 ab	0.39 b	0.55 a	0.33 a	0.34 b	0.26 ac	0.29 c	0.35 b	0.51 a	0.36 b	0.40 b
Ile	0.54 ab	0.69 ab	0.51 b	0.77 a	0.33 bc	0.48 a	0.32 c	0.37 b	0.58 c	1.62 a	0.57 c	1.09 b
Leu	0.86	1.08	0.84	1.29	0.30 b	0.49 a	0.29 b	0.35 b	0.57 c	1.50 a	0.54 c	0.97 b
Lys	0.45	0.63	0.42	0.84	0.06 c	0.21 a	0.05 c	0.12b	--	--	--	--
Phe	0.68 ab	0.77 ab	0.59 b	0.90 a	0.36 c	0.49 ab	0.37 bc	0.43 a	0.54 c	1.26 a	0.53 c	0.90 b

Data are the means of 6 replicates. The different letters indicate significant differences (*p* < 0.05).

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
