# Peer review of "Physiological, Hormonal and Metabolic Responses of two Alfalfa Cultivars with Contrasting Responses to Drought"

_ijms, 2019, doi:10.3390/ijms20205099_

Round 1
Reviewer 1 Report
The topic of the manuscript is interesting and as a whole is well presented. However, there are some adjustments to be made to improve the manuscript.
Too many References have been included in this manuscript. It is not a Review.
Accurate control of English throughout the manuscript in recommended.
I agree on that water stress led to a significant repression of the photosynthetic rate, transpiration and stomatal conductance. However, the effect of water stress on Ci concentration seems to be not significant. In addition, the repressive effect of water stress on A, E and gs is not the same in the two cultivars. Indeed, the San Isidro cultivar shows a more negative effect than Zhong Mu one. When you commented the results on 2iP you wrote “The small differences between treatments and varieties was significant” did you mean that the differences observed for leaves were significant? The results reported in Figure 5 show that the IAA levels also decreased in leaves of San Isidro cultivar. There are no comments on the results concerning the amino acids analysis. The statistical method used to indicate the significance of the data should be also introduced in the Table notes. This comment is also valid for all the Figures reported in the manuscript. 228-229: Two contrasting concept have been expressed in these lines. Please re-write these sentences. 267-268: Check this sentence. Probably the correct sentence is: “In contrast, jasmonic acid was strongly reduced in nodules under drought with the greater reduction observed for San Isidro”. 291: …”the greater decrease of…”. 306: Please change this sentence. “….the larger increase in root glucose…”. 320: Please change this sentence. “….the increased levels registered in drought-stressed plants….”. 324-326: Please check this sentence. The expressed concept is not clear. 330-331: Please change this sentence. “Asn is the main….in alfalfa, and Asn accumulation in nodules has been implicated in… N2 fixation”. 334-341: Please make this part more fluid. 436-439: Please change this sentence. “According to….conservative response: reducing of water loss by closing stomata and fixing less carbon by photosynthesis, thus limiting its grwth under water limitation”. 447-449: Please make this part more fluid. Too many References have been included in this manuscript. It is not a Review. Accurate control of English throughout the manuscript in recommended.
Reviewer 2 Report
The presented manuscript comprehensively evaluates drought stress response in two "cultivars" of alfalfa. The studied varieties were wisely chosen to present different way of reacting to water deficit. It allowed to indicate differential responses in part of the measured parameters that could suggest a genetic background of such differences. Also, the significant plasticity of reactions as suggested in the varied physiological responses. Therefore, the topic and scope of the study is sound and interesting to plant science and is also well fit to this special issue (…
The applied methodology is limited to physiological and developmental response without any attempt of looking into the gene expression level of response. It is a pity but does not diminish the value of the study in terms of insight into the physiology. The physiological part is well designed and sufficient to draw conclusions in the Discussion. There are some corrections necessary which are listed further below.
Also, I would have a general suggestion to make the writing more concise by avoiding too many general statements about water stress and response and focusing directly on the studied phenomena. Now, the discussion is thorough but rather long and wordy. Please, try to delete or shorten what you would find not absolutely necessary to understand the context.
Here are more specific comments and suggestion to improve the paper:
line36-37 – why only from equator to Siberia – what about Southern hemisphere?
lines 55-72 – this paragraph can be easily shortened;
lines 92-95 – this sentence is expected to provide clear aim of the study and it doesn't. It only states what was done to "elucidate the key features" – actually, quite vague and difficult to get convinced by that.
Figures (bar graphs) 2-3 – add the legend boxes of both varieties (like in fig. 1), not just ww and wd. And use consistent terms- either well watered or control;
further, most of the important comments refer to Materials and Methods, so please, run through them carefully:
The origin and identity of plant material must be exactly mentioned - source of plants (what does it mean from Spain/China?), accession number, or vendor batch etc; How was it verified that these are indeed "genotypes"? If genotypes weren't actually verified, avoid using this term; Hormone quantification – you refer to the paper by Müller & Munné-Bosch. However, it is unclear if you've used exactly the same configuration and parameters as theey, or there were some deviations (e.g column make – there – AMT Halo was used which isn't so popular, in fact; etc). Amino acid quantification – the same question as above. Sugars quantification – only the chromatograph was mentioned – details are needed for the readers to be able to repeat the method. Or give the sufficient reference complemented with additional information if applicable; C and N -the same as above;
Author Response
Comments:
The discussion is thorough but rather long and wordy… Following the comment made by the referee, the discussion has been shortened in order to highlight the findings described in the paper.
Introduction
Why only from equator to Siberia (Lines 36-37). This sentence has been corrected. This paragraph can be easily shortened. (Lines 55-72). As suggested, we have shortened this paragraph in order to make it clearer. This sentence is expected to provide clear aim of the study and it doesn’t. (Lines 92-95). In agreement with the comment made, we have modified the word “elucidate” for “reveal”. Figures 2-3 – add the legend boxes of both varieties (like in fig. 1), The legend boxes have been added to the figures 2 and 3.Materials and Methods
The origin and identity of plant material must be exactly mentioned…How was it verified that these are indeed “genotypes”? Both alfalfa cultivars (San Isidro and Zhong Mu) are commercial and widely cultivated cultivars in Spain and China respectively. In order to avoid potential misunderstandings, we have changed the term “genotype” for “cultivar”. 6. Hormone quantification – you refer to the paper by Müller & Munné-Bosch. However, it is unclear if you’ve used exactly the same configuration and parameters as they, or there were some deviations… More details of the extraction process and the model of the UHPLC have been added in order to make it more understandable. Amino acid quantification – the same as above. Sugars quantification – only the chromatograph was mentioned… As suggested, additional information has been given on the protocol followed to determine amino acids and sugars contents. C and N – the same as above. Similarly, a more detailed information of the method used to quantify C and N contents has been given in the new version of the manuscript.Reviewer 3 Report
The manuscript (Physiological, hormonal and metabolic responses of two alfalfa cultivars with contrasting responses to drought) describes changes in the content of some sugars, amino acids and selected phytohormones in 2 alfalfa cultivars growing under drought stress.This manuscript is relatively well written and I can recomend this manuscript for publication in IJMS after revisions listed below:
Abstract needs to be improved. Most of the text is about the results of your previous study, not about the current article. The materials and methods section should be more comprehensive. Details regarding experimental design should be added: especially the chapter 4.3. Hormone profiling: missing sample weight, source of internal standards, the type of used UHPLC-MS/MS Did you use FW or DW for hormone profiling, from methodology is obvious using of FW but in the results you stated DW, what is right……..? Some abbreviations are not listed.
Author Response
Comments:
Abstract need to be improved. The abstract has been revised following the recommendation made by the referee. The Material and Methods section should be more comprehensive. Especially the chapter 4.3 Hormone profiling: missing sample weight, source of internal standards, type of used UHPLC-MS/MS. In agreement with the comment made, additional information on the protocol used have been added to the text (sample weight, source of internal standards, and type of used UHPLC-MS/MS). Did you use FW or DW for hormone profiling, from methodology is obvious using of FW but in the results you stated DW, what is right…..? The extraction processes was conducted in fresh material. But, in order to avoid potential differences on the determinations linked with the hydration level, the results were expressed on dry weigh basis. This sentence has been added in the results to avoid any confusion.4. Some abbreviations are not listed. The abbreviations in the text have been revised accordingly.